# Neurohormonal Modulation as a Therapeutic Target in Pulmonary Hypertension

**DOI:** 10.3390/cells9112521

**Published:** 2020-11-22

**Authors:** Inés García-Lunar, Daniel Pereda, Borja Ibanez, Ana García-Álvarez

**Affiliations:** 1Centro Nacional de Investigaciones Cardiovasculares Carlos III (CNIC), 28029 Madrid, Spain; ines.garcia@cnic.es (I.G.-L.); bibanez@cnic.es (B.I.); 2Hospital Universitario Quirónsalud Madrid, UEM, 28223 Madrid, Spain; 3Centro de Investigación Biomédica en Red (CIBER) de Enfermedades Cardiovasculares, 28029 Madrid, Spain; dpereda@clinic.cat; 4Institut Clínic Cardiovascular-Hospital Clínic, IDIBAPS, Universitat de Barcelona, 08036 Barcelona, Spain; 5Department of Cardiology, IIS-Fundación Jiménez Díaz, 28040 Madrid, Spain

**Keywords:** adrenoceptor signaling, sympathetic regulation, pulmonary hypertension, animal models, denervation

## Abstract

The autonomic nervous system (ANS) and renin-angiotensin-aldosterone system (RAAS) are involved in many cardiovascular disorders, including pulmonary hypertension (PH). The current review focuses on the role of the ANS and RAAS activation in PH and updated evidence of potential therapies targeting both systems in this condition, particularly in Groups 1 and 2. State of the art knowledge in preclinical and clinical use of pharmacologic drugs (beta-blockers, beta-three adrenoceptor agonists, or renin-angiotensin-aldosterone signaling drugs) and invasive procedures, such as pulmonary artery denervation, is provided.

## 1. Introduction

Pulmonary hypertension (PH) is a clinical syndrome characterized by an increase in pulmonary artery pressure (PAP) and pulmonary vascular resistance (PVR), associated with progressive right ventricular (RV) dysfunction and poor prognosis. According to physiopathological, clinical, and therapeutic considerations, PH is divided into the following five groups: Group 1, pulmonary arterial hypertension (PAH); Group 2, PH secondary to heart failure (HF); Group 3, PH secondary to lung disease or hypoxia; Group 4, PH secondary to pulmonary artery obstruction; and Group 5, PH of unclear/multifactorial mechanism [1].

The autonomic nervous system (ANS) and renin-angiotensin-aldosterone system (RAAS) are central to the neurohormonal regulation of cardiovascular and pulmonary function and are closely involved in many cardiopulmonary disorders. In this review, we focus on the role of neurohormonal activation in PH and current evidence of potential therapies targeting the ANS/RAAS in this condition, particularly in Groups 1 and 2. PAH, although infrequent (prevalence of 16 cases/million [2]), is the hallmark of PH and the group in which the most important advances in the understanding and treatment of the disease have been achieved during the last decade. Still, current treatments for PAH remain limited to relaxing the pulmonary vasculature, aiming to reduce RV afterload, and secondarily improving RV function, but have limited direct effects on cardiac function or arterial remodeling. Although pulmonary vasodilator drugs (i.e., prostanoids, endothelin receptor antagonists, phosphodiesterase-5 inhibitors, and soluble guanylate cyclase stimulators) have demonstrated benefit in improving exercise capacity and slightly prolong survival of PAH patients, they do not provide a substantial increase in lifespan or quality of life. Group 2 PH is more prevalent than Group 1 with HF being the most common etiology of PH globally. It is estimated that approximately 50% of patients with chronic HF have secondary PH [3], which results in more severe symptoms, worse exercise tolerance, and poorer prognosis [4,5]. Despite this burden of disease, unfortunately there are no specific pharmacological treatments available for patients with Group 2 PH [6]. Thus, further research on the mechanisms involved and novel therapeutic targets with robust prognostic impact are necessary for both PAH and PH secondary to HF, the latter even of greater importance since it could be applied to a large, rapidly increasing number of patients.

The pulmonary vasculature is innervated by sympathetic, parasympathetic, and sensory nerve fibers [7]. The distribution of nerves around the PA has been studied in large animals [8,9] and humans [10]. In humans, sympathetic and parasympathetic nerves arising from the spinal ganglions and the vagus nerve, respectively, merge to form the anterior and posterior plexi at the carina. From here, nerves enter the lungs forming a peribronchial plexus and a periarterial plexus; the latter runs within the adventitial layer and innervates the pulmonary vasculature, reaching up to small PAs of <100 μm diameter [11] (Figure 1). Dedicated immunohistochemistry for tyrosine hydroxylase has demonstrated that the predominant innervation is sympathetic (70% of fibers are positive) [10].

## 2. Involvement of the Autonomic Nervous System in Pulmonary Hypertension (PH)

There is increasing evidence linking the ANS with the pathogenesis of PH, specifically in PAH [7,12,13,14]. This activation has been confirmed using direct (plasmatic catecholamine concentration) [14], as well as indirect techniques (muscle sympathetic nerve activity, heart rate variability and baroreflex sensitivity) [12,13,15]. However, circulating plasma catecholamines have been found to be increased in some studies and normal in others [12,16,17]. These discrepancies might be explained by the fact that plasma catecholamines are affected by neuronal release, reuptake, spillover, and degradation, and therefore are insensitive markers of sympathetic nerve outflow in pathological conditions [12]. There is also evidence from small series showing cardiac sympathetic involvement in PAH using nuclear imaging [18]. Heart rate variability, baroreflex sensitivity parameters, and muscle sympathetic nerve activity have been further related to a poor functional capacity and clinical worsening in PAH [12,13,19,20].

Closely related, the RAAS is upregulated in PAH and has been associated with clinical worsening [7,21]. Interestingly, an increase of angiotensin II receptor AT1 but not AT2 has been described in the pulmonary vasculature of PAH patients [21], which has important implications for potential treatments since the binding of angiotensin II to its AT1 and AT2 receptors has opposite effects, i.e., AT1 signaling is involved in vasoconstriction, oxidative stress, inflammation and proliferation while AT2 signaling results in vasodilation. Although the initial trigger of neurohormonal activation in PAH has not been fully elucidated, it has been proposed that pulmonary vascular damage, increased RV wall stress, and renal hypoperfusion may upregulate the local and systemic RAAS [11,22]. As occurs in HF, in which overactivation of the sympathetic nervous system and RAAS is one of the key elements of the disease [23], this activation initially may act as a compensatory response to a drop in cardiac output, but might have a detrimental effect in the long term if maintained chronically. Indeed, chronic upregulation of neurohormones results in downregulation of the β1-adrenergic receptor, impairing the inotropic and lusitropic responsiveness of the heart and eventually leading to severe RV dysfunction.

## 3. Pharmacological Strategies Targeting the Autonomic Nervous System in PH

### 3.1. Beta-Blockers

Some authors have hypothesized that treatment with β-blockers could be useful in patients with PH and RV dysfunction, similar to the treatment of left HF. On the contrary, others disagree based on the idea that patients with PH are highly heart rate-dependent to maintain and increase cardiac output. As we will see below, there is also a discrepancy between the results of preclinical experimental studies (most favorable) and clinical reports (most showing no benefit).

Preclinical data coming from rat models mimicking PAH have found a beneficial effect of β-blocker treatment in several pathways, including a decrease in cardiomyocyte hypertrophy, fibrosis, capillary rarefaction, ischemia, apoptosis, oxidative stress, and inflammation [24,25,26,27], associated with an increase in RV function. In this sense, Ishikawa et al. [24] showed that arotinolol, a pure α/β-adrenoceptor antagonist, prevented PH development and RV hypertrophy in monocrotaline (MCT)-exposed rats. Subsequently, Bogaard et al. [25] and later Okumura et al. [28] reported that carvedilol, a non-selective β-blocker, improved RV function despite persistent pressure overload in the experimental rat PH models of Sugen/hypoxia and MCT, respectively. In the study by Bogaard et al., the beneficial effect observed with carvedilol on RV adaptation was superior to that of metoprolol, a selective β1-blocker, which was interpreted as driven by α1-adrenoreceptor blockade and/or antioxidant/anti-inflammatory properties associated with carvedilol. However, using pressure-volume loop analysis, de Man et al. [26] described a load-independent beneficial effect of another selective β1-blocker, bisoprolol, on the RV in the MCT rat model. Finally, Perros et al. [27] compared the effect of the two selective β-blockers nebivolol and metoprolol on cultures of human pulmonary artery (PA) endothelial cells, pulmonary arterial rings and biventricular function in rats with MCT-induced PH. Only nebivolol significantly decreased human pulmonary endothelial cell proliferation and improved nitric oxide (NO)-dependent relaxation and the effect of nebivolol on pulmonary hemodynamics and RV performance in vivo was greater than metoprolol. Interestingly, nebivolol is a β1-adrenergic receptor antagonist but has an associated β2/3-adrenergic receptor agonist activity which confers vasodilator properties and possibly other effects associated with β3-agonism, as we will discuss later.

Despite these promising results with the use of β-blockers in the experimental field, its clinical use in PAH remains controversial. Clinical studies, though mostly small sample-sized and non-randomized, have not shown a clear benefit with some of them reporting systemic hypotension and decreased exercise capacity, believed to be secondary to their negative inotropic and chronotropic effect [29,30,31,32,33,34]. The first evidence regarding their potential deleterious effect came from a series of cases of patients with porto-PH in which β-blockers (atenolol or propranolol) withdrawal resulted in improved functional capacity and cardiac output [31,32]. Later, the common use of β-blockers in PAH patients, due to the presence of concomitant cardiac comorbidities (particularly supraventricular arrhythmias), has allowed continued investigations of their potential clinical effect. In a prospective analysis of a cohort of 94 PAH patients divided according to the use of β-blockers at baseline, there were no differences in pulmonary hemodynamics, RV performance, or morbimortality, after a median follow-up of 20 months [35]. Moreover, another observational single-center prospective study [36] that followed PH patients (most of them with PAH) according to the use of β-blockers (bisoprolol as the most frequent choice) found a reduction in RV diameter and improvement in tricuspid annular plane systolic excursion (TAPSE) associated with a reduction in natriuretic peptides. Obviously, these results should be interpreted with caution due to the nature of the study (lack of randomization and statistical techniques to balance population characteristics). In a retrospective propensity-matched analysis from 508 PAH patients, Bandyopadhyay et al. [29] found no differences in survival and time to clinical worsening, although β-blocker use (metoprolol as the most commonly prescribed) was associated with a tendency towards shorter walking distance. Comparable neutral impact in all-cause mortality was found in another propensity-matched analysis of a large PAH cohort [33] with the use of β-blockers (again metoprolol as the most frequently used). In a small pilot study [30], including six PAH patients with stable condition but RV dysfunction, focused on the evaluation of RV performance using cardiac magnetic resonance (CMR), six months after starting treatment with carvedilol (median tolerated dose of 18.75 mg twice daily), the authors reported a significant improvement in RV ejection fraction, although difficult to interpret because B-type peptide natriuretic (BNP) levels increased and there were no differences in exercise capacity. Similarly, in a small randomized placebo-controlled clinical trial [37], an improvement in RV systolic function as assessed by echocardiography and positron emission tomography-measured glucose uptake was observed, with no changes in RV cardiac output nor exercise capacity. In contrast, in a randomized placebo-controlled crossover clinical trial including 18 patients with idiopathic PAH, patients treated with bisoprolol (until a maximum tolerated dose of 10 mg/daily) did not show a significant improvement in RV ejection fraction (main outcome) but suffered a significant deterioration of cardiac output and a near significant decrease in a six-minute walking test (6MWT) [34]. Trying to explain these disappointing results of clinical studies, Rijnierse et al. [38] recently reported that bisoprolol did not modify RV sympathetic activity quantified by positron emission tomography or RV function. In summary, there is no robust evidence of the beneficial effect of β-blockers in PAH; the use of different types and doses of β-blockers and patient characteristics might explain the controversial results reported in the literature. Table 1 shows the details of the main characteristics and findings of preclinical and clinical studies performed with β-blockers in PH. In the light of these data, current guidelines advise against the use of β-blockers in PAH, unless required by other comorbidities, such as hypertension, coronary artery disease, or left HF [39].

In Group 2 PH, treatment with β-blockers is even less clear, since this entity includes patients with HF with preserved and reduced left ventricular ejection fraction (LVEF), as well as valvular heart disease. While β-blockers are one of the cornerstones of treatment in patients with reduced LVEF they have demonstrated no benefit in the other two contexts [40]. In addition, the incidence of PH in randomized controlled trials with β-blockers in patients with HF with either reduced or preserved LVEF has not been specifically addressed [41]. There is currently a registered trial aimed to evaluate the potential beneficial effect of nebivolol on hemodynamics and 6MWT in PH secondary to HF with preserved LVEF (NCT02053246).

### 3.2. Beta-3 Adrenergic Receptor Agonists

In the last decades, there has been growing evidence for the implication of beta-three adrenergic receptor (β3-AR) in the pathophysiology of cardiovascular diseases. β3-AR mRNA expression has been found in the human myocardium [42] and vessels [43], and it has been described to be upregulated in left heart disease [42]. The coupling of β3-AR to the nitric oxide (NO)/cyclic guanosine monophosphate (cGMP) pathway results in a cardiovascular effect opposed to classical β1- and β2-AR effects, that it is thought to protect against deleterious adrenergic overactivation [44]. In recent years, several publications have demonstrated the cardioprotective effects of β3-AR stimulation in different experimental models of ischemia-reperfusion injury [45,46,47] and HF [48,49], including HF with preserved LVEF [50], through a NO-mediated mechanism. In 2017, Bundgaard et al. [51] published the first-in-man randomized clinical trial using mirabegron, a β3-AR agonist, in patients with HF. Although there were no differences in the main outcome (LVEF), treatment was well-tolerated and showed no safety concerns. In this line, there are two ongoing clinical trials to evaluate the therapeutic benefit and safety of β3-AR agonist in patients with HF with preserved LVEF (NCT 02599480) [52] and reduced LVEF (NCT01876433). Regarding the potential effect of β3-AR on the pulmonary vasculature, there was preliminary evidence showing the vasodilatory effect of ex vivo β3-AR stimulation in pulmonary vessels from dogs [53] and rats [54]. Specifically focused on PH, our group reported a beneficial effect of β3-AR agonist treatment on pulmonary hemodynamics and RV performance in a randomized placebo-controlled study in an experimental model of post-capillary PH in pigs [55]. Animals receiving mirabegron showed significantly lower PVR and higher RV ejection fraction (as assessed by CMR) and cardiac output at follow-up. These functional effects were associated with changes in protein expression suggestive of attenuated vascular proliferation in the lung parenchyma. In addition, β3-AR was found to be expressed in human pulmonary arteries and β3-AR agonist administration inhibited human pulmonary smooth muscle cell proliferation in vitro by a NO-dependent mechanism and produced vasodilation of human pulmonary arterial rings in organ bath studies. A clinical trial is currently recruiting patients to assess the therapeutic potential and safety of mirabegron in Group 2 PH patients (NCT02775539) [56]. The β3-AR agonist properties of the β1-AR blocker nebivolol might explain its incremental beneficial effect reported in experimental PH as compared with other β-blockers [27,57]. Conversely, a very recent experimental study has reported a potential beneficial effect of the β3-AR antagonist SR59230 in RV function with reduction of inflammatory infiltration to the lung in a model of PH induced by MCT in rats [58]. However, the absence of pulmonary hemodynamic evaluation in this study (and therefore the confirmation of PH) and the particularities of the MCT model (systemic toxicity and inflammation) may explain the contradictory results and question the benefit of β3-AR antagonists in PH patients. Ongoing trials will hopefully clarify the role of β3-ARs in the management of PAH.

## 4. Pharmacologic Therapies Targeting the Renin-Angiotensin-Aldosterone System in PH

According to the described involvement of RAAS signaling in PH, the following two different strategies have been tested: First, the inhibition of the angiotensin-converting enzyme (ACE) with drugs like enalapril or the inhibition of the angiotensin II AT1 receptor with losartan; and secondly, the stimulation of the ACE2/angiotensin-(1-7) axis, considered vasoprotective [59]. Table 2 depicts the main characteristics and findings of preclinical and clinical studies performed with drugs targeting the RAAS in PH.

In initial experiments, enalapril was shown to inhibit the development of PH and RV hypertrophy in MCT rats [60] by a NO and p21-dependent mechanism; and by tumoral necrosis factor (TNF) inhibition in bleomycin-exposed mice [61]. However, with the exception of few small series of cases [73,74], there has not been robust clinical evidence associated with the benefit of enalapril or any other ACE inhibitors in any group of PH. There is also conflicting evidence regarding the effect of the angiotensin-1 receptor antagonist losartan in experimental PH and a lack of clear clinical benefits. Whereas losartan significantly reduced PH progression, RV dilatation and pulmonary vascular remodeling in MCT-induced PH rats in the study by de Man et al. [21], it did not shown any prophylactic effect in the same model in prior studies by Cassis et al. [62] or Kreutz et al. [75]. Similar neutral effects were observed with the use of losartan (either in combination with eplerenone or as compared with bisoprolol) in a rat model of right HF due to pressure overload [63]. However, in the particular scenario of experimental shunt-induced PH, Rondelet et al. [64] reported a preventive effect of losartan associated with a reduction in BMPR-2 expression. Regarding the stimulation of the ACE2/angiotensin-(1–7) axis, although a positive effect has been repeatedly reported in the experimental field in MCT-induced PH [65,66,67,68], there is limited data confirming safety and efficacy of these drugs in PH patients. In the proof-of-concept pilot trial by Hennes et al. [69], five PAH patients received a single intravenous (i.v.) dose of recombinant human ACE2 without significant safety concerns, but a very heterogenous response in PVR. Therefore, further clinical evidence with these compounds is required. Finally, aldosterone is another key player of the RAAS. Maron et al. [70] demonstrated that spironolactone or eplerenone, mineralocorticoid receptor antagonists, prevented or reversed pulmonary vascular remodeling and improved cardiopulmonary hemodynamics in MCT and Sugen/hypoxia-induced PH models in rat. Similar results were obtained by Preston et al. [71] in two small animal models of PH (hypoxia on mice and MCT on rats). Despite these promising results, there is a paucity of data confirming the potential beneficial effect of these drugs on the clinical arena. In this line, Maron et al performed a subanalysis of patients in whom spironolactone use was reported in the ARIES-1 and -2 studies (in combination to ambrisentan) and compared them with patients with ambrisentan alone and observed that the former had a trend towards reduced BNP levels and improvement in 6MWT and WHO functional class [72]. A randomized placebo-controlled clinical trial is currently ongoing to evaluate the potential benefit and safety of spironolactone in early phases of PAH, before the development of RV failure [76] (NCT01712620).

## 5. Invasive Strategies Targeting the Autonomic Nervous System in PH

### 5.1. Pulmonary Artery Denervation

In experimental studies performed in the 80s, balloon distension of the main PA was found to increase PAP and PVR and this increase was abolished by surgical denervation of the PA bifurcation, as well as by chemical sympathectomy, thus, indicating that the efferent branch of this reflex is predominantly mediated by the sympathetic nervous system [77]. Thirty years later, Chen et al. [78] suggested, for the first time, that percutaneous PA denervation (using a dedicated catheter) could acutely reduce the increase in PA pressure and PVR generated by a balloon occlusion of a pulmonary branch. Since this first publication, PA denervation has been further tested in another seven experimental studies [8,9,10,79,80,81,82], the details of their main characteristics and findings can be found in Table 3. Studies using acute models of PH have reported an acute improvement in PA hemodynamics after the percutaneous PA denervation procedures using radiofrequency [78] or intravascular ultrasound [10]. However, these results are difficult to extrapolate to the human disease, since PH was secondary to acute occlusion or vasoconstriction and not to progressive pulmonary vascular remodeling, as in patients with chronic PH.

Studies based on chronic models of PH have mainly used the injection of MCT [9,79,82] that causes pulmonary vascular remodeling, increased pulmonary pressures and subsequent RV overload, trying to mimic the pathophysiology of PAH, but also induce a general multi-organ inflammatory response, which is not present in patients with PAH, and therefore limiting the translation of results to the clinical practice. Despite the fact that the model is not ideal to represent human PAH, there are two studies with this approach that have found a positive effect of catheter-based PA denervation in pulmonary hemodynamics, as well as RV and pulmonary vascular remodeling [9,79]. In 2019, the study by Hang et al. [80] used a different method to generate PH by performing a surgical banding of the ascending aorta in rats and a combined technique (surgical microdissection of the PA and phenol application) for the PA denervation procedures. Although the study was again positive in terms of PA hemodynamics, RV function, and pulmonary vascular remodeling, the low efficacy and high mortality of the PH model (78 rats were required to have 18 animals with complete follow-up) together with the probable variability of manual dissection of the PA adventitia in a small animal, made the results difficult to interpret. The first negative experimental study in PA denervation was published in 2019 by Garcia-Lunar et al. [81] and used a different methodology from prior publications. First, the PH model was a swine model of chronic postcapillary PH (generated by a pulmonary vein banding) and the authors used a surgical approach with application of radiofrequency directly to the external layer of the PA with bipolar clamps for the PA denervation with histological confirmation. Transmural denervation, in this study, produced no benefit in terms of PA hemodynamics and a trend towards increased biventricular volumes and RV mass evaluated with CMR and histology. There are two plausible reasons for the apparent discrepancies with previous publications. First, complete PA denervation in this specific model with high postcapillary pressure could hypothetically have worsened pulmonary congestion, thus, resulting in negative effects as previously observed with other pulmonary vasodilators when applied to Group 2 PH patients. Second, a disbalanced sympathetic/parasympathetic denervation favoring the latter might have a deleterious effect on RV remodeling. In a second arm of the same study, the authors found on histological analysis that the damage produced on the PA and both branches was incomplete when they used an intravascular catheter to perform the procedures, unlike what happened with the surgical clamps in the first arm. Later, in 2019, Huang et al. [82] published an experimental study in rats with chronic MCT-induced PH, in which surgical denervation (by manual dissection of the tissue surrounding the main PA and both branches plus placement of a patch to avoid reinnervation) was associated with better hemodynamics, pulmonary vascular remodeling, and downregulation of the sympathetic nervous system and RAAS. They also reported lower ventricular volumes and higher RV ejection fraction in denervated animals (using CMR), but they only performed one study per animal (post procedure), therefore it was impossible to know if groups were homogeneous before randomization. In addition, the authors did not remove the adventitia of the PA because they had results from a pilot study (including 13 rats and 3 patients) in which they found a high density of nervous fibers in the connective and adipose tissue surrounding the PA bifurcation. However, ANS fibers have been previously recognized to be found within the adventitia of the vessels [11,93]. Moreover, if these fibers were indeed localized in the periadventitial tissue, as Huang et al. suggested, it would be feasible to obtain a successful denervation using surgical dissection or external clamps, but less probably using a monopolar catheter inside the vessel lumen.

In the clinical area, we have data from five studies on PA denervation in patients with PH, only two of them being randomized trials [85,86], see Table 3. Briefly, shortly after their first experimental proof, in 2013, Chen et al. directly applied percutaneous PA denervation to a group of 13 idiopathic PAH patients [83], which was further extended to 66 patients with PH of varied etiology (39 PAH, 18 Group 2 PH and nine operated chronic thromboembolic PH (CTEPH) [84]. The results were positive in terms of hemodynamic parameters, functional class, and RV function measured with echocardiography up to one year after the procedures. In 2019, Zhang et al. published the results of PADN-5, the only randomized controlled trial (although not blinded) in Group 2 PH available so far [85]. This trial included patients with combined pre- and postcapillary PH (CpcPH), an entity that has been associated with reduced exercise capacity and a histological phenotype similar to PAH [94]. In PADN-5, percutaneous PA denervation was associated with an improvement in 6MWT distance and a reduction in PVR as compared with sildenafil treatment plus a control procedure. However, the results are difficult to interpret since we have solid scientific evidence warning against the use of sildenafil in Group 2 PH [95]. Systemic hypotension in the control arm may have precluded the uptitration of HF-specific medications, thus, accounting for a worse evolution in this group. In addition, very recently, results from the TROPHY-1 study (a multicenter international early feasibility trial) have been published [87]. In this study, PA denervation using an intravascular ultrasound device was performed in 23 patients with PAH on dual or triple oral therapy and not responsive to acute vasodilator test. There were no procedure-related serious adverse events reported and a total of seven procedure-related adverse events (described as bleeding, hematoma, bruising, or low saturation). In the 20 patients that underwent functional and hemodynamic assessment at follow-up, there was a significant reduction in PVR and increase in 6MWT distance six months after the intervention. Surprisingly, hemodynamic changes were not present acutely after the PADN procedures but developed afterwards, which leaves the question open about what is the real mechanism by which PADN may exert a beneficial effect. Over the 12-month follow-up period, there was one death (unrelated to PADN) and 11 PAH-related hospitalizations, which is considered to be an acceptable rate given the characteristics of the study population. Finally, although beyond the scope of this review, early in 2020, positive results from a randomized, single-blind clinical trial of PADN vs. sham procedure plus medical therapy with riociguat in CTEPH patients have been reported [86].

In summary, we have some preliminary positive data suggesting that percutaneous PA denervation could be an effective therapy for chronic PH, with more convincing data for PAH. Nevertheless, there are important gaps of knowledge regarding the mechanism of action and potential benefit of this intervention. For example, how a percutaneous denervation that does not fully interrupt all sympathetic nervous fibers [81] may have a predictable and reproducible effect on pulmonary hemodynamics or if there is a way to avoid injuring parasympathetic nerve fibers during the procedure. Until now, PADN trials have used empirical strategies to guide the denervation procedures, mainly based on fluoroscopic landmarks [11]. Some authors have proposed to use mapping of the ANS using nuclear medicine techniques [96] to optimize PADN procedures and one case reported on the use of high-output burst electrical stimulation to map the ablation sites on computed tomography images [97]. Similarly, in a very recent experimental trial, ablation at certain PA sites with evoked heart rate responses, lead to disappearance of neural markers’ expression [98]. In the CTEPH randomized clinical trial, a three-dimensional electroanatomical mapping system was used with remote magnetic navigation for the PADN procedures for the first time [86]. In any case, as long as the catheter applications are solely based on where sympathetic nerve fibers should lay (which is based, in turn, on anatomic descriptions from three to four cases [10,82]), it appears that the efficacy of the denervation procedures will largely depend on individual distribution and depth of the sympathetic/parasympathetic fibers and the size, thickness, and composition of the PA wall. This limitation may be overcome by the recently described PA mapping strategies [86,98].

Additionally, it seems plausible that sympathetic denervation may reduce the component of functional vasoconstriction associated with chronic PH, but which are the mechanisms involved in the reduction in vascular remodeling from small pulmonary vessels remains to be elucidated.

Furthermore, one would assume that a successful sympathetic denervation would reduce systemic blood pressure and heart rate [99]; however, in most clinical and experimental studies these parameters remain unchanged [10,85]. Regarding the durability of the intervention, it has also not been clearly established whether there may be a role for autonomic reinnervation, as was suggested by Huang et al. [82] and has already been reported in heart transplant recipients and swine renal arteries [100]. If this were the case, the intervention may lose efficacy over time and re-do procedures may be necessary. Finally, studies with dedicated RV anatomy and function evaluation are needed to elucidate whether there may be a direct and potentially deleterious effect of PA denervation on cardiac remodeling and function [81].

All these uncertainties are especially relevant in the subgroup of patients with Group 2 PH, where we have no convincing evidence of the effectiveness of the intervention. Despite this, percutaneous PA denervation is already being applied to patients with PH secondary to left heart disease [80,83,84,85,97,101]. Given the uncertainty that still many experts share about the effectiveness of PA denervation as a treatment for chronic PH [11,102,103], there is a need for randomized, controlled, multicenter blinded studies in this field [104,105]. Blinding is particularly important because the placebo effect has proven to be very relevant in previous renal denervation trials [106]. Currently there are two ongoing event-driven trials on PA denervation in PAH (PADN-PAH, NCT 02,284,737 and PADN-CDFA, NCT 03282266) and a pilot trial on ultrasound PA denervation for patients with left heart disease (TROPHY-II, NCT 03611270).

### 5.2. Renal Denervation as a Treatment for PH

Catheter-based renal denervation is an interventional procedure aiming to interrupt the afferent and efferent sympathetic renal innervation by targeting the renal adventitial nerve plexus with an endovascular catheter. After the positive results of the open-label SYMPLICITY-2 clinical trial on the treatment of arterial hypertension [107], renal denervation has been applied to other cardiovascular diseases (such as HF or atrial fibrillation) with the intention to reduce the underlying neurohormonal activation. In 2015, Quingyan et al. [88] used, for the first time, this approach in MCT-induced PAH in dogs. In their study, surgical renal denervation (used as a prevention strategy, that is, immediately after the MCT injection and before PAH had been stablished) was able to ameliorate pulmonary hemodynamics, RV and pulmonary vascular remodeling, and decrease plasmatic and pulmonary RAAS levels. Though innovative, the results should be interpreted cautiously given that the translation of results to human PH is limited (because of the model and the experimental design) and that the control group did not undergo a sham intervention [108]. Liu et al. also tested surgical renal denervation in a rat MCT model of PH as a preventive/treatment strategy and found that early denervation was associated with better pulmonary vascular and RV remodeling [89]. Later on, Da Silva Gonçalves Bos [90] performed renal denervation in a study involving two experimental rat models of chronic PH, i.e., MCT and Sugen/hypoxia. They found that renal denervation was able to reduce PVR as well as pulmonary vascular remodeling, RV hypertrophy, and fibrosis. On pressure-volume loop evaluation, renal denervation significantly reduced RV afterload (Ea) and diastolic stiffness (Eed). However, there was no significant change in RV systolic pressure, cardiac output, or RV size and systolic function (TAPSE and end-systolic elastance (Ees)), which again leaves unanswered how to explain the cardiovascular improvement after the procedure.

### 5.3. Sympathetic Ganglion Block

Sympathetic ganglion block by injection of a local anesthetic agent has been used as a treatment for patients with ventricular arrhythmias [109]. This approach has also been investigated as a therapy in an experimental study in rats with MCT-induced PAH [91]. This study found that RV systolic pressures were lower in the animals that underwent sympathetic ganglion block as compared with sham controls (who received saline injections instead). This was associated with decreased RV hypertrophy and pulmonary vascular remodeling. Sympathetic ganglion block was also able to reduce oxidative stress and to suppress arginase activity, increasing NO availability, thus, further strengthening the possible crosstalk between the ANS and NO that had been previously postulated. More recently, Zhao et al. [92] used transection of the cervical sympathetic trunk (a procedure leading to long-term sympathetic ganglion block) in MCT rats and found that the procedure was associated with a decrease in noradrenaline concentration in the lung, reduced RV systolic pressure, RV and pulmonary vascular remodeling, and increased RV performance.

### 5.4. Parasympathetic Stimulation

The modulation of the parasympathetic system is complex and has been less used as a target in cardiovascular diseases, including PH, as compared with the inhibition of the sympathetic system. Da Silva Gonçalves Bos et al. [110] reported that pyridostigmine, an oral drug stimulating the parasympathetic activity through acetylcholinesterase inhibition, reduced PVR, RV hypertrophy and dilatation, and pulmonary vascular remodeling in Sugen/hypoxia rats. These effects were associated with a reduction in fibrosis and inflammation in the myocardium and lung parenchyma.

Electric vagal nerve stimulation (VNS) has been postulated as a potential therapy in several cardiovascular diseases such as ventricular arrhythmias or chronic HF, although its efficacy remains controversial [111]. Yoshida et al. [112] evaluated the potential effect on survival and related mechanisms of preventive or therapeutic VNS in the experimental PH model of Sugen/hypoxia in rats. They found that VNS markedly improved the survival rate in PH rats. In the pathophysiologic study, they showed a significant reduction in plasma norepinephrine, pulmonary hemodynamics, pulmonary vascular remodeling, and increased cardiac output. Although the results were promising, the study had important limitations, that included the difficulty in assessing mortality in experimental research, the significant percentage of failure to perform the VNS, and the evident differences between the Sugen/hypoxia model and human PAH.

## 6. Conclusions

The role of neurohormonal activation has gained attention in the past years as a pathophysiological driver and possible treatment target for chronic PH. The current review summarizes the existing evidence on pharmacological, as well as invasive strategies, targeting the ANS/RAAS in PH. Although the scope of the review is PH from Groups 1 and 2, we acknowledge that a large proportion of the experimental and clinical evidence presented here refers to animal models mimicking PAH (mainly MCT in small animals) or patients with PAH. Further experimental and clinical research efforts in Group 2 PH would be highly relevant given the paucity of data in this field and the growing number of patients with PH secondary to HF.

Several experimental studies have suggested a beneficial role of pharmacologic neurohormonal inhibitors (such as β-blockers, ACE inhibitors, angiotensin-receptor blockers, ACE2-activators, or spironolactone) in PH. However, clinical studies (though mostly small sample sized and not controlled) have failed to demonstrate benefit, thus precluding the use of these drugs as established treatments for PH patients. Stimulation of the β3-AR has also proven beneficial in experimental HF and PH, and currently there is a phase-2 clinical trial on CpcPH patients on the way. There has also been increasing interest in invasive strategies targeting the ANS, particularly percutaneous PA denervation. This intervention has shown early positive signs in experimental and non-randomized clinical studies but questions on its mechanism of action, potential benefit, and long-term efficacy of the procedure remain to be addressed. Ongoing and future randomized, controlled, multicenter, and blinded clinical trials should reveal whether this strategy may become part of the routine clinical care for chronic PH patients.

## Figures and Tables

**Figure 1 cells-09-02521-f001:**
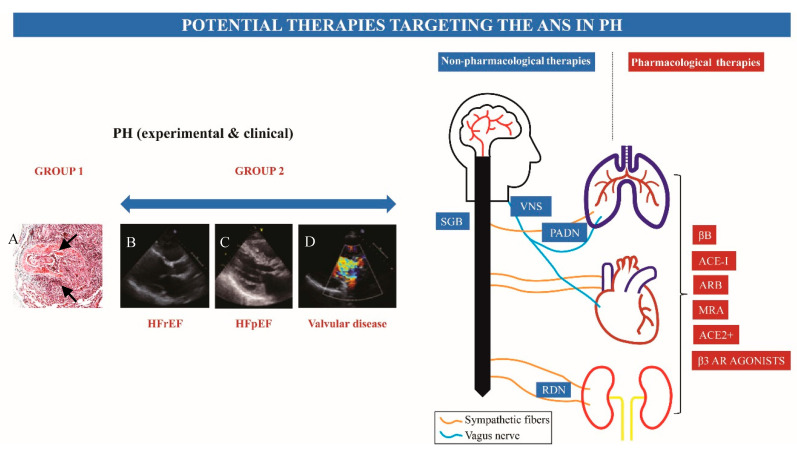
Schematic figure summarizing neurohormonal activation in pulmonary hypertension (PH) and pharmacologic and invasive treatments targeting the autonomic nervous system (ANS)/ renin-angiotensin-aldosterone system (RAAS) in this condition. (**A**) Microphotograph of pulmonary vascular remodeling in experimental pulmonary arterial hypertension (PAH) secondary to systemic-to-pulmonary shunt (in piglets). Black arrows point to a plexiform lesion, one of the histopathological hallmarks of pulmonary arterial hypertension (PAH); (**B**–**D**). Figures of parasternal long-axis transthoracic echocardiograms illustrating the spectrum of patients with PH secondary to heart failure (HF). (**B**) A patient with HF with reduced ejection fraction (secondary to dilated cardiomyopathy): (**C**) HF with preserved ejection fraction (secondary to hypertrophic cardiomyopathy); and (**D**) HF secondary to valvular heart disease (severe aortic regurgitation). ACE-I, angiotensin-converting enzyme inhibitors; ACE2+, angiotensin-converting enzyme stimulator; AR, adrenergic receptor; ARB, angiotensin receptor blocker; βB, beta-blocker; HFpEF, heart failure with preserved ejection fraction; HFrEF, heart failure with reduced ejection fraction; MRA, mineralocorticoid receptor antagonist; PADN, pulmonary artery denervation; RDN, renal denervation; SGB, sympathetic ganglion block; VNS, vagal nerve stimulation.

**Table 1 cells-09-02521-t001:** Beta-blockers in PH.

Author	Experimental (Model) or Clinical (Patients)	Treatment	Study Design	N	Results
Ishikawa [24]	Experimental (MCT rats)	Arotinolol	Controlled by saline-treated animals (2 weeks, preventive strategy)	N = 6 per group	↓ pulmonary vascular remodeling ↓ mPAP ↓ RV hypertrophy (RV/BW ratio)
Bogaard [25]	Experimental (Sugen/hypoxia rats)	Carvedilol (and metoprolol)	Controlled by saline-treated animals (initiated 2 weeks after MCT)	N = 12 per group	↓ RV hypertrophy (RV weight) ↓ RV dysfunction ↓ RV fibrosis, apoptosis and capillary rarefaction
De Man [26]	Experimental (MCT rats)	Bisoprolol	Controlled by saline-treated animals (initiated 10 days after MCT)	N = 7 per group	↑ RV contractility ↑ RV CO ↓ RV fibrosis
Perros [27]	Experimental (MCT rats)	Nebivolol (and metoprolol)	Controlled by saline-treated animals (initiated 2 weeks after MCT, for 1 week)	N = 10 per group	↓ PVR ↓ RV CO ↓ RV hypertrophy
Okumura [28]	Experimental (MCT rats)	Carvedilol	Controlled by saline-treated animals (initiated 2 weeks after MCT)	N = 7 (saline) and N = 8 (carvedilol)	↓ RV dysfunction ↓ RV hypertrophy ↓ RV fibrosis
Bandyopadhyay [29]	Clinical (iPAH, associated PAH)	Atenolol, bisoprolol, carvedilol, metoprolol, nebivolol, propranolol, sotalol	Observational retrospective propensity-matched analysis	N = 508	No significant differences in survival or time to clinical worsening.
Grinnan [30]	Clinical (iPAH, hPAH, aPAH)	Carvedilol	Single-arm pilot open-label	N = 6	↑ RV ejection fraction ↑ BNP No differences in exercise capacity
Thenappan [33]	Clinical (iPAH, hPAH, aPAH, drug induced PAH)	Atenolol, carvedilol, labetalol, metoprolol, nadolol, propranolol	Observational retrospective propensity-matched analysis	N = 564	No differences in all-cause mortality
Van Campen [34]	Clinical (iPAH)	Bisoprolol	Randomized placebo-controlled trial with crossover design	N = 18	No differences in RV ejection fraction ↓ RV CO ↓ 6MWT distance (trend)
So [35]	Clinical (iPAH, associated PAH, drug-induced PAH)	Acebutonol, atenolol, bisoprolol, metoprolol, nadolol, propranolol	Observational (prospective cohort)	N = 94	No differences in hemodynamic, RV performance, 6MWT distance, hospitalizations or all-cause mortality
Moretti [36]	Clinical (iPAH, aPAH, CTEPH, pre and post-capillary PH, and others.	Bisoprolol, atenolol, metoprolol, nadolol, propranolol.	Observational (prospective cohort)	N = 94	↑ TAPSE ↓ RV diameter
Farha [37]	Clinical (PAH, hPAH, aPAH, PH due to lung disease, CTEPH)	Carvedilol	Randomized placebo-controlled trial	N = 30	↑ RV fractional area change ↓ RV glucose uptake ↓ RV systolic pressure

aPAH, associated PAH; BNP, B-type natriuretic peptide; CO, cardiac output; CTEPH, chronic thromboembolic pulmonary hypertension; hPAH, hereditary PAH; iPAH, idiopathic PAH; MCT, monocrotaline; mPAP, mean pulmonary artery pressure; PVR, pulmonary vascular resistance; RV, right ventricular; PAH, pulmonary arterial hypertension; PH, pulmonary hypertension; TAPSE, tricuspid annular plane systolic excursion; 6MWT, 6-minute walking test.

**Table 2 cells-09-02521-t002:** Drugs targeting RAAS in PH.

Author	Experimental (Model) vs. Clinical (Patients)	Treatment	Study Design	N	Results
Kanno [60]	Experimental (MCT rats)	Enalapril	Controlled by saline-treated animals (5 weeks, preventive strategy)	N = 8 per group	↓ RV hypertrophy (weight and CMR-measured).
Ortiz [61]	Experimental (blemycin-treated mice)	Enalapril	Controlled by saline-treated animals (2 weeks)	N = 9 per group	↓ Pulmonary hemodynamics ↓ RV hypertrophy (free wall weight)
De Man [21]	Experimental (MCT rats)	Losartan	Controlled by vehicle-treated animals (initiated 10 days after MCT, for a maximum of 25 days)	N = 9 per group	↓ SPAP estimated by echo. ↓ RV dilatation ↓ Pulmonary vascular remodeling
Cassis [62]	Experimental (MCT rats)	Losartan	Controlled by saline-treated animals (20 days, preventive strategy)	N = 10 per group	No differences in SPAP or RV hypertrophy. No differences in medial pulmonary arterial thickening.
Borgdorff [63]	Experimental (PA banding in rats)	Losartan (plus eplerenone)	Controlled by saline-treated animals (preventive strategy)	N = 15 per group	No differences in RV pressure-volume loops. No differences in RV performance.
Rondelet [64]	Experimental (systemic to pulmonary shunt in piglets)	Losartan	Randomized placebo-controlled study (preventive strategy, for 3 months)	N = 8 (Losartan) vs. N = 10 (placebo)	↓ mPAP, PVR ↓ RV dP/dtmax
Ferreira [65]	MCT rats	ACE2-activator	Controlled by saline-treated animals (preventive strategy)	N = 13 vs. n = 7 (placebo)	↓ RVSP ↓ RV hypertrophy (RV/LV+S ratio) ↓ PA medial wall thickness ↑ IL10
Bruce [66]	MCT rats	ACE2-activator	Controlled by vehicle-treated animals (initiated 2 weeks after MCT, for a maximum of 4 weeks)	N = 14 per group.	↓ RVSP ↓RVEDP & dP/dt_max_ ↓ Interstitial and perivascular fibrosis ↓ RV fibrosis
Shenoy [67]	MCT rats	ACE2-activator	Controlled by vehicle-treated animals (both preventive and therapeutic strategies)	N = 6 to 8 per group.	↓ RVSP ↓ RVEDP & dP/dt_max_ ↓ PA medial Wall thickness ↓ RV fibrosis
Li [68]	MCT + pneumonectomy rats	ACE2-activator (resorcinolnaphthalein)	Controlled by saline-treated animals (preventive strategy)	N = 8 per group	↓ mPAP ↓ RV hypertrophy (RV/LV+S ratio) ↓ PA neointimal formation
Hemnes [69]	PAH patients	ACE2-activator (GSK2586881)	Pilot single-arm study	N = 5	No statistically significant changes in hemodynamics, or biventricular performance by echocardiography
Maron [70]	MCT rats and Sugen/hypoxia rats	Spironolactone or eplerenone	Controlled by vehicle-treated animals (both preventive and therapeutic strategies)	N = 3 to 4 per group	↓ Pulmonary vascular remodeling ↓ PVR ↓ REDOX generation and restored ET-dependent NO production
Preston [71]	Hypoxia mice and MCT rats	Spironolactone	Controlled by vehicle-treated animals (both preventive and therapeutic strategies)	N = 6 to 8 per group	↓ PVR ↓ /~Pulmonary vascular remodeling No differences in RV hypertrophy (Fulton index) ↓ RV fibrosis
Maron [72]	PAH patients	Spironolactone (+ ambrisentan) vs. ambrisentan alone	Retrospective subanalysis of randomized placebo-controlled trials ARIES-1 and 2	N = 31 vs. 57 (ambrisentan alone)	Trend towards: ↓ BNP leve l ↓ WHO class ↓ 6MWT distance

BNP, B-type natriuretic peptide; CMR, cardiac magnetic resonance;; ET, endothelium; LV, Left ventricular; MCT, monocrotaline; mPAP, mean pulmonary arterial pressure; NO, nitric oxide; PAH, pulmonary arterial hypertension; PH, pulmonary hypertension; PVR, pulmonary vascular resistance; RAAS, renin-angiotensin-aldosterone system; RV, right ventricular; RVEDP, RV end-diastolic pressure; RVSP, RV systolic pressure; S, septum; SPAP, systolic pulmonary arterial pressure; WHO, World Health Organization; 6MWT, 6-minute walking test.

**Table 3 cells-09-02521-t003:** Invasive strategies targeting the ANS in PH.

Author	Experimental (Model) vs. Clinical (Patients)	Treatment	Study Design	N	Results
Chen [78]	Experimental (balloon occlusion dogs)	PADN with RF catheter	Pre-post analysis (no control group)	20	After PADN: mPAP, PVR, and CO remain stable after balloon occlusion of the PA branch
Rothman [8]	Experimental (TxA2 infusion pigs)	PADN with RF catheter	Controlled by sham procedure	8	After PADN: the TxA2 infusion produces a dampened response in PAP, PVR, and CO in PADN animals
Zhou [9]	Experimental (MCT dogs)	PADN with RF catheter	Controlled by sham procedure (8 weeks after MCT)	20	↓ PAP, ↓ PVR, ↑ CO, ↓ RV hypertrophy and pulmonary vascular remodeling, ↓ SNS conduction velocity, demyelinization and axon loss
Liu [79]	Experimental (MCT dogs)	PADN with RF catheter	Controlled by sham procedure (8 weeks after MCT)	16	↓ PAP, ↓ PVR. ↓ RV hypertrophy and pulmonary vascular remodeling. ↓ RAAS activation.
Hang [80]	Experimental (aortic banding rats) + clinical (Group 2 PH)	Exp: PADN (Surgical + chemical) Clin: PADN (RF catheter)	Exp: Controlled by sham procedure (4 weeks after aortic banding) Clin: Not controlled	Exp: 13 Clin: 10	Exp: ↓ right atrial pressure, ↓ RV systolic pressure, ↓ RV hypertrophy, ↑ RV function, ↓ pulmonary vascular remodeling, change in adrenoreceptor concentration in the lungs Clin: ↓ PAP, ↓ PVR, ↑ CO, ↑ 6-minute walking distance, no change in RV size/function
Garcia-Lunar [81]	Experimental (chronic postcapillary PH pigs)	PADN with surgical RF ablation clamps	Controlled by sham procedure (2 months after PV banding)	12	No change in hemodynamic parameters or pulmonary vascular remodeling, trend towards ↑ biventricular volumes and RV mass (by histology and CMR), ↓ NA levels at the RV
Huang [82]	Experimental (MCT rats)	PADN (surgical)	Controlled by sham procedure (4 weeks after MCT)	20	↓ PAP, ↓ RV systolic pressure, ↓ RV hypertrophy and fibrosis, ↓ pulmonary vascular remodeling, ↓ SNS and RAAS activation
Chen [83]	Clinical (idiopathic PAH)	PADN with RF catheter	Single-center open-label trial (cases vs. controls who refused PADN)	21	↓ PAP, ↑ 6MWT distance, ↑ RV function (Tei index) Safety: Chest pain during the procedure.
Chen [84]	Clinical (varied etiology, Groups 1, 2 and 4).	PADN with RF catheter	Single-center non-controlled trial	66	↓ PAP, ↓ right atrial pressure, ↓ RV systolic pressure, ↑ CO, ↓ PVR, ↑ 6MWT distance and functional class, ↓ NT-proBNP, ↓ size and ↑ RV function (Tei index) Safety: Chest pain during the procedure, 8 deaths, and 18 rehospitalizations within 12 months.
Zhang [85]	Clinical (CpcPH 39% HFpEF, 61% HFrEF)	PADN with RF catheter	Multicenter randomized open-label trial, controlled by sham + sildenafil treatment	98	↓ PAP, PCWP, PVR, diastolic gradient, and ↑ CO, ↑ LV and RV function (Tei index), ↑ 6MWT distance ↓ clinical worsening (post-hoc analysis) Safety: No report of procedural complications, 7 deaths, and 2 pulmonary embolisms within 6 months.
Romanov [86]	Clinical (CTEPH)	PADN with RF catheter	Single center single-blind randomized trial, controlled by sham + riociguat treatment	50	↓ PAP, PVR, ↑ 6MWT distance Safety: Chest pain and cough during the procedure, 32% of patients developed bradycardia or transient asystole that was resolved with temporary pacing through the ablation catheter
Rothman [87]	Clinical (PAH on dual or triple therapy)	PADN with US catheter	Multicenter open-label trial	23	↓ PAP, PVR, right atrial pressure, ↑ 6MWT distance Safety: No procedure-related serious adverse events, 1 death, and 11 rehospitalizations within the 12-month follow-up
Qingyan [88]	Experimental (MCT dogs)	Renal denervation with RF catheter	Not controlled (cases vs. PAH dogs) Preventive strategy	22	↓ PAP, PVR, ↓ RV and pulmonary vascular remodeling, ↓ neurohormonal activation
Liu [89]	Experimental (MCT rats)	Renal denervation (surgical + fenol)	Controlled by sham surgery (24 h and 2 weeks after MCT)	40	No hemodynamic evaluation, ↓ pulmonary vascular remodeling and RV fibrosis, ↓ sympathetic nerve system and RAAS activity, early renal denervation (preventive strategy) was associated with better remodeling
Da Silva Gonçalves Bos [90]	Experimental (MCT and Su-Hx rats)	Renal denervation (surgical + fenol)	Controlled by sham surgery (2 or 6 weeks after MCT/Su-Hx)	38	↓ PVR, no significant change in RV systolic pressure/CO, ↓ pulmonary vascular remodeling, RV fibrosis, and hypertrophy, ↓ RAAS (pulmonary vasculature/RV)
Na [91]	Experimental (MCT rats)	Sympathetic ganglion block	Controlled by sham (saline injections) starting 2 weeks after MCT	20	↓ RV systolic pressure, ↓ pulmonary vascular remodeling, RV fibrosis, and hypertrophy, ↓ oxidative stress, ↑ NO availability
Zhao [92]	Experimental (MCT rats)	Transection of cervical sympathetic trunk	Controlled by sham surgery Preventive strategy	26	↓ RV systolic pressure, ↓ RV and pulmonary vascular remodeling, ↑ RV function, ↓ lung levels of NA

CO, cardiac output; Clin, clinical; CTEPH, chronic thromboembolic pulmonary hypertension; Exp, experimental; MCT, monocrotaline; NA, noradrenaline; NT-proBNP, N-terminal prohormone of brain natriuretic peptide; LV, left ventricle; PA, pulmonary artery; PADN, pulmonary artery denervation; PAP, pulmonary artery pressure; PCWP, pulmonary capillary wedge pressure; PVR, pulmonary vascular resistance; RAAS, renin-angiotensin-aldosteron system; RF, radiofrequency; RV, right ventricle; SNS, sympathetic nervous system; Su-Hx, Sugen-hypoxia; TxA2, thromboxan A2; US, ultrasound; 6MWT, Six-minute walking test.

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
