# Peer review of "Neurohormonal Modulation as a Therapeutic Target in Pulmonary Hypertension"

_cells, 2020, doi:10.3390/cells9112521_

Round 1

Reviewer 1 Report

Pulmonary arterial hypertension is a progressive disease with high mortality. Although, current approved therapies showed improvements in quality of life and hemodynamic parameters, they have demonstrated only very limited beneficial effects on survival and disease progression. Thus, there is a pressing unmet need for more effective treatments of pulmonary arterial hypertension. Accumulating evidence suggests the role of autonomous nervous system

This is an excellent review of the current knowledge related to the involvement of autonomous nervous system and renin-angiotensin-aldosterone system in pulmonary hypertension and treatment strategies targeting both systems. The work is a very nice addition to the field and covers an area that is often overlooked. This review requires only very minor modifications before publication.

Minor

As the renin-angiotensin-aldosterone system does not belong to the autonomous nervous system, I would suggest to modify the title of the manuscript.

In order to be more specific, I would suggest to write “pulmonary vascular remodeling” instead of “lung remodeling” (lines 454 and 455).

Reviewer 2 Report

This is a well written, in-depth review article summarizing ANS targeted potential therapies in PH.  It is well organized with pathobiology clearly explained prior to reviewing experimental and clinical trial potential therapies. This review article thoughtfully captures the most up to date data along with their limitations as well as provides theories for discrepant or contradictory findings. It conveys a promising outlook for ANS targeted therapies, particularly with PADN, and offers insight on work that needs to be done to advance this discipline of therapies.

One noted minor limitation: The primary aim of this review article was to focus on the role of potential ANS targeted therapies in group 1 and group 2 patients. The experimental and clinical studies for all mechanisms (BB, RAAS, PADN), however, predominantly studied WHO group 1 PAH or the animal model equivalent (MCT rats) phenotype. This appears to be a limitation in the field more so than of the article but I think warrants mentioning.

Only minor suggestions as below:

Line 50- Demonstrated to improve – Demonstrated benefit in improving

Line 52- HF should be defined as heart failure then abbreviated. That sentence is confusing. Can be modified to: “Group 2 PH is more prevalent than group 1 with HF being the most common etiology”

Figure 1- Particularly, histopathology photo and Echo features of group1 and group 2 PH- There is no legend explanation for these images and I don’t feel these help support the accompanying figure of ANS innervation.

Line 165- Modified can be changed to Modify

Line 201- Vasodilate can be changed to vasodilatory

Line 325-Short can be changed to shortly

Line 383- associated to can be changed to associate with
